# Spatial Distribution of Snow Cover in Tibet and Topographic Dependence

**Duo Chu [1,2,\*], Linshan Liu [3] and Zhaofeng Wang [3]**

1   Tibet Institute of Plateau Atmospheric and Environmental Sciences, Tibet Meteorological Bureau, Lhasa 850000, China
2   Tibet Key Laboratory of Plateau Atmosphere and Environment Research, Science and Technology Department of Tibet Autonomous Region, Lhasa 850000, China
3   Key Laboratory of Land Surface Pattern and Simulation, Institute of Geographic Sciences and Natural Resources Research, CAS, Beijing 100101, China; liuls@igsnrr.ac.cn (L.L.); wangzf@igsnrr.ac.cn (Z.W.)
\*   Correspondence: chu_d22@hotmail.com; Tel.: +86-138-8908-2802

**Abstract:** Many major river systems in Asia, such as the Yangtze, Yarlung Zangbo, Indus, Ganges and Salween originate in the Tibetan mountains and snow cover in Tibet provides substantial water resources for these rivers, in addition to its weather-related and climatic significance. The high mountain terrain of Tibet is the main condition that snow cover exists and persists at mid–low altitudes. However, the relationships between snow cover and topographic factors of the plateau have not been fully addressed. In this study, the overall spatial distribution of snow cover and the impacts of topography (elevation, aspect and slope) on snow cover distribution in Tibet were analyzed based on the MODIS snow cover product and digital elevation model (DEM) using GIS spatial analysis techniques. The results showed that (1) snow cover in Tibet is spatially very uneven and is characterized by rich snow and high SCF (snow cover frequency) on Nyainqentanglha mountain and the surrounding high mountains, with less snow and a low SCF in the southern Tibetan valley and central part of northern Tibet. (2) Snow cover in Tibet has a strong elevation dependence and a higher SCF corresponds well with high mountain ranges. The mean SCF below 2000 m above sea level (m a.s.l) was less than 4%, while above 6000 m a.s.l, it reached 75%. (3) Intra-annual snow cover distribution below 4000 m a.s.l was characterized by unimodal patterns, while above 4000 m a.s.l, it was characterized by bimodal patterns. The lowest SCF below 6000 m a.s.l occurred in summer, while above 6000 m it occurred in winter. (4) The mountain slope and aspect affect snow cover distribution through changing radiation and energy balances in the mountain regions. The mean SCF generally increased with mountain slopes, with the highest on the north-facing aspect and the lowest on the south-facing aspect.

**Keywords:** snow cover frequency; topographic elements; MODIS data; SRTM DEM; Tibet





## 1. Introduction

The cryosphere is defined by the presence of frozen water in its many forms: glaciers, snow, permafrost, ice caps, and river and lake ice [1], playing an important role in the hydrological cycle. The cryosphere is an integral element of high mountain regions, which are home to roughly 10% of the global population [2]. Snow cover, as a key component of the cryosphere, is a sensitive indicator of climate change and also affects the climate system through changing the energy balance and water cycle between land and atmosphere due to its high albedo and hydrological effect. Moreover, as the natural solid water reservoir of the cryosphere, snow cover provides more than one-sixth of the world's population with fresh drinking water through snow melt, and further supports other uses such as hydropower generation, irrigation and industry [3–8]. The Tibetan Plateau (TP) is the highest and largest mountain region on Earth. The TP and surrounding high mountains are known as the "third pole of the world" since this region has the largest global store of frozen water after

the polar regions and functions as the Asian water tower [9,10]. Snow cover on the TP is a unique surface feature in the mid–low latitudes of the Northern Hemisphere (NH) and exerts an important influence on regional and large-scale weather and climate systems in the TP and beyond [11–13]. On the other hand, the snow-covered area on the TP is the headwaters for many large rivers in the western China and Himalayan regions. The ice- and snow-melt water provides important water resources for these river systems and regulates seasonal water supply and river runoff, which is essential for the lives of people and their livelihood in the plateau and downstream areas [12–14].

Snow cover in the mountain region is not only closely related to climatic conditions (temperature and precipitation) but also varies with topography (elevation, aspect, and slope) [15,16]. However, it is difficult to obtain accurate information on the large-scale snow cover distribution and topographic impacts on the snow cover on the TP based on the current sparse ground observation network. Satellite remote sensing with a large spatial coverage, and various spatial and temporal resolutions, coupled with digital elevation model (DEM) data, makes it possible to reveal the spatial distribution of snow cover and topographic impacts on snow cover in the mountain regions. The high mountain topography of the TP is the main condition for the presence and accumulation of snow on the surface and largely controls the spatial distribution and temporal variations of snow cover on the plateau. On the TP, snow cover can persist at higher-altitude regions during all seasons and varies with the terrain features, such as elevation, slope and aspect [15]. Furthermore, snow cover persistence and duration generally become longer with increases in the terrain elevation [17,18]. The higher snow cover frequency (SCF) is mostly concentrated in the regions where the elevation is higher than 6000 m. Snow cover on the TP depends on the elevation. Snow cover at lower altitudes reaches the maximum in January; at above 6000 m altitude, the average snow coverage in the four seasons is greater than 60% and there are two peaks within the year, in spring and autumn [15]. The length of the snow-covered season on the TP appears to be decreasing at lower elevations because of the increase in air temperature. However, at higher elevations, the increase in precipitation appears to compensate for the increase in air temperature such that the snow-free period has decreased [19].

Snow cover changes in the Hindu Kush Himalaya (HKH) regions are highly variable because of various types of controlling factors, such as topographic effects, glacier dynamics and geomorphological parameters, and they generally show altitudinal dependence [20–24]. Snow cover distribution in the Himalayans is largely controlled by latitude and altitude, and generally, the elevation of the terrain plays a decisive role in snow accumulation [21]. The change in snow cover with altitude clearly shows a sharp jump in terms of percentage snow cover between 5001 and 6400 m of elevation. A comprehensive spatiotemporal analysis of snow phenology for the Tienshan Mountains shows that the spatiotemporal changes of snow phenology have strong altitude dependence [16,25–28]. Due to differences in solar radiation and water vapor sources, the north-facing areas generally have a higher snow cover duration (SCD), earlier snow onset date (SOD), and later snow end date (SED) than south-facing areas. In addition to elevation, aspect also can affect snow phenology by altering local solar radiation and moisture conditions, as well as potentially the accumulation regime, due to windward/leeward effects [29]. Snow cover change in the global mountain region is closely related to altitude. Air temperature is the main driver of snow onset and melt, while a combined effect of air temperature and precipitation dominates in the winter season [30–33].

Tibet here refers to the Tibet Autonomous Region (TAR) administratively in China and is the main body of the TP. Snow cover in Tibet constitutes the main snow cover on the TP and is a seasonally fast changing land cover feature in Tibet. Like the TP region, the high mountain topography of Tibet is the main condition for the presence and persistence of snow cover in this mid–low region of the NH. However, how the mountain topography of Tibet affects the spatial distribution of snow cover has not been fully investigated so far. There has been a lack of quantitative analysis between snow cover and topographic factors in the Tibet area. In this study, the overall spatial distribution of snow cover in Tibet was

examined first. Subsequently, the impacts of topographic elements (elevation, slope and aspect) on the snow cover distribution in Tibet were analyzed in depth based on the MODIS snow cover data in combination with DEM data using the GIS spatial analysis method. This study is of great significance for understanding the spatial snow cover characteristics, snow water resource availability and regional differences, as well as regional water resource management and use.

## 2. Study Area

Tibet, also referred to as TAR, is the short name for TAR and is located in the southwestern part of the TP, which is shown in Figure 1. Tibet (TAR) stretches around 2000 km from west to east at 78°24′–99°06′ E and 1000 km from south to north at 26°52′–36°32′ N. It covers an area of more than 1.2 million square kilometers, accounting for approximately one-eighth of China's total land area. Tibet is the main part of the TP, the highest mountain region in the world, with an average altitude of over 4000 m a.s.l, commonly known as "the roof of the world".

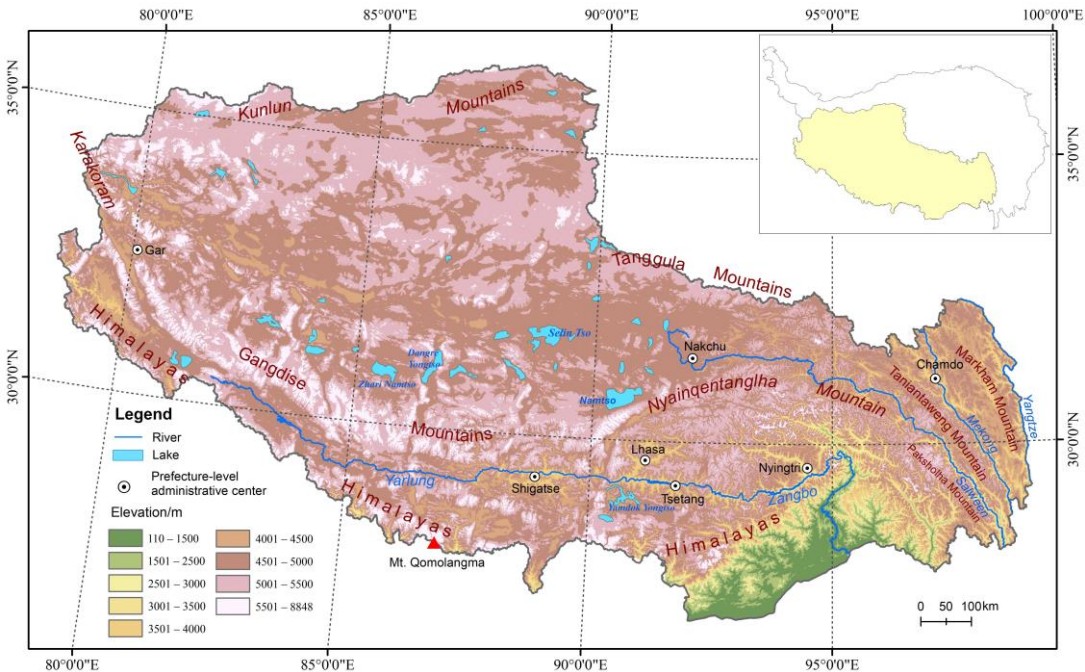

**Figure 1.** Main mountain ranges, rivers, lakes and topography in Tibet.

Tibet is dominated by westerlies in winter and by Indian monsoon in summer. Spring and autumn are transition seasons between winter and summer atmospheric circulations. The climate in Tibet is generally characterized by alpine climate with marked regional differences. The annual mean temperature is −2.2–12.2 °C and decreases from south to north. Annual precipitation ranges from 70 to less than 900 mm with a decreasing pattern from southeast to northwest. Annual precipitation in most of Tibet is below 400 mm and is primarily concentrated in the rainy season from May to September, accounting for 84% of the annual total precipitation [34].

Tibet is surrounded by high mountains, and valleys and deep rivers lie in the south and east, while the north is vast Qiangtang plateau and the central Tibet is interspersed with mountains, valleys and lake basins [35,36]. The landform in Tibet is diverse and complex with distinct regional differences. It is generally categorized into the southern Himalayan mountains, the lake and basin area in northern Himalayas, the central Yarlung Zangbo river valley, the northern plateau and lake basin area, and the southeastern alpine gorge region [37].

## 3. Data and Methods

### 3.1. Remote Sensing Data

In the study, snow cover data used were MOD10A2 snow cover product v005 from 2001 to 2014, downloaded from the U.S. National Snow and Ice Data Center. MOD10A2 is 8-day composite data of MODIS daily snow cover products (MOD10A1) through the 8-day maximum composite method to eliminate cloud obscuration, and provides more consistent and cloud-free coverage than daily products. The spatial resolution is 500 m and sinusoidal projection is used for the products. According to a tile of area of $1200 \times 1200$ km$^2$, the world is divided into 18 rows and 36 columns in sinusoidal projection, with a total of 648 tiles. The coordinates of tiles start from h00v00. MODIS product v005 is generated based on the previous product v004 using improved algorithms.

According to the spectral response characteristics of different land cover features, snow cover on the surface has high reflection in the visible band and strong absorption in the shortwave infrared band, whereas cloud generally has high reflectance in both the visible and near-infrared bands. Based on the spectral signature of snow cover and cloud, the normalized differential snow cover index (NDSI) is used for the MODIS snow cover mapping method to identify snow cover on the surface [38,39].

In this method, if NDSI $\geq$ 0.40, the reflectance of the near-infrared band 2 > 0.11 and the reflectance of band 4 > 0.1, the pixel is classified as snow. Moreover, NDSI is not sensitive to the large-scale illumination condition and reduces the atmospheric effect through normalization and not depending on a single band reflection value. Since the Terra satellite's revisit period is 16 days and the ground revisit period is 8 days, the best surface coverage from the Terra satellite can be obtained by selecting the 8-day composite method for eliminating cloud contamination. In MOD10A2 data, a pixel is considered as snow if snow appeared at least once during the 8-day period and cloud if cloud cover is observed on all 8 days.

Since MODIS data are available to the public, the accuracy of the MODIS snow mapping algorithm has been validated worldwide. In the NH, the annual mean error of MODIS snow cover products under clear sky conditions is around 8% [40]. In the northwest of China, the overall accuracy of MODIS daily snow product MOD10A1 under clear sky is over 90%, and MOD10A2 products can effectively eliminate the influence of cloud on snow cover mapping on the surface [41–43]. Pu et al. first evaluated the accuracy of MOD10A2 snow cover product on the TP by comparing the data with snow observations from meteorological stations, and the results showed that the total accuracy of the product was about 90% [15]. It is evident that MOD10A2 snow cover data have a high accuracy for snow cover mapping to adequately capture the spatial variability of snow cover in mountain regions of western China and the TP area.

The study area includes 6 tiles of MOD10A2, namely, h24v05, h24v06, h25v05, h25v06, h26v05 and h26v06. The pixel values in MOD10A2 data were labeled as 25, 37, 50 and 200, representing snow-free, lake, cloud and snow, respectively. There were a total of 686 images from March 2000 to February 2015, among which 14 years of images from January 2001 to December 2014 were used for annual analysis and 15 years of images from March 2000 to February 2015 were used for seasonal analysis. The procedure of MODIS image processing is all MOD10A2 is mosaiced and converted from HDF files into GeoTIFF files first using MRT (MODIS Reprojection Tools); secondly, the sinusoidal projection is transformed into geographic coordinates using MRT tools with the nearest neighbor resampling method and WGS84 as ellipsoid; thirdly, the GeoTIFF files are converted into ArcGIS GRID format; finally, output image is clipped to a subset of the study area for further spatial analysis.

### 3.2. DEM Data

The DEM (Digital Elevation Model) data used are SRTM (Shuttle Radar Topography Mission) DEM data archived in the U.S. Geological Survey's Center for Earth Resources Observations and Science (USGS EROS), and the spatial resolution is 90 m. DEM data are then resampled to a 500 m spatial resolution to be consistent with the MOD10A2 product

and is transformed from a GeoTIFF format to an ArcGIS GRID format. It was reported that the absolute vertical accuracy of these data was greater than 16 m and absolute horizontal accuracy was greater than 20 m [44]. Huang et al. evaluated the accuracy of SRTM DEM data in the TP area by comparing it with ICESat laser altimetry data, and it was found that the elevation difference is $1.03 \pm 15.20$ m, which is higher than the original designing accuracy of 16 m [45].

Based on the DEM data, the elevation, slope and aspect data were generated and zoned according to specific topographic characteristics and implications in the Tibet area. The elevation was categorized into 7 zones with 1 km interval, namely, below 1 km, 1–2 km, 2–3 km, 3–4 km, 4–5 km, 5–6 km, and above 6 km of elevation. The slope was categorized into four zones, namely, below 5°, 5–10°, 10–20°, and above 20°. Likewise, the aspect was categorized into the north-facing aspect (315–45°), east-facing aspect (45–135°), south-facing aspect (135–225°), and west-facing aspect (225–315°) at an interval of 90° in a clockwise direction with starting from 315°, corresponding to shady slope, semi-sunny slope, sunny slope and semi-shady slope for mountain terrains, respectively. No aspect region (i.e., flat terrain) was represented by "−1".

*3.3. Snow Cover Frequency*

To examine snow cover persistence and occurrence in the study area, snow cover frequency (SCF) was defined to characterize the percentage of snow cover pixels in total pixels in MOD10A2 time-series data. The equation was expressed as follows:

$$SCF = \left[ \frac{1}{M} \sum_{i=1}^{M} \sum_{k=1}^{D} \frac{N_{ik(1)}}{D} \right] \times 100\% \tag{1}$$

where, *SCF* is the snow cover frequency as a percentage. $N_{ik}$ is the pixel value of surface feature in the *k*th image of MOD10A2 time-series data in *i*th year. For the snow cover pixel, the value is set to 1, while for none-snow cover pixel, the value is set to 0. *M* is the total number of years that MOD10A2 data are available. *D* is the total number of MOD10A2 images in a year or season. The four seasons were spring (March–April–May), summer (June–July–August), autumn (September–October–November) and winter (December–January and February of the next year).

## 4. Results and Discussion

*4.1. Spatial Distribution of Snow Cover*

4.1.1. Annual Mean SCF

The spatial distribution of annual mean snow cover in Tibet was mapped using Equation (1), and the result is shown in Figure 2. Snow cover on the plateau is very unevenly distributed and generally presents that there is rich snow and more perennial snow cover with high SCF in the southeastern and surrounding high mountain regions, whereas there is less snow and a low SCF in the southern Tibetan valley and the central part of northern Tibet. Snow cover in Tibet is strongly associated with terrain features, with a longer duration in the high mountains and shorter duration in the vast interior and river valleys. The higher SCF corresponds well with the high mountain ranges. Annual mean SCF from 2001 to 2014 was 16.3%, and the area with mean SCF < 10% accounts for around half of total area of the plateau and is mainly distributed in the upper and middle Yarlung Zangbo river valleys, central part of northern Tibet, and river valleys in the southeast. The area with mean SCF < 20% is 72.3% of the plateau area, while the area with mean SCF < 50% is 93.4% of total plateau area. If a mean SCF of 60% is considered as the threshold for seasonal and perennial snow cover, seasonal snow cover in Tibet accounts for 96.4% while perennial snow cover is only 3.6%. Therefore, most of snow cover in Tibet is seasonal, and areas of perennial snow cover are quite limited and principally distributed in the high mountain ranges, such as Nyainqentanglha mountain and its southeastern extension, the Kunlun and Tanggula mountains in the north, western Gangdise mountain, and the

Himalayas in the south. At higher elevation in the mountain regions, lower air temperature and more precipitation create favorable conditions for the formation and maintenance of snow cover on the surface, whereas due to strong shielding from these huge mountains, snow cover is relatively scarcer in the most interior regions of the TP and southern Tibetan river valleys [33,46]. Previous study shows that snow cover distribution on the TP is far from a pervasive feature and that of peripheral mountains is appreciable snow cover, while in the vast interior snow cover is limited and of a short duration [18].

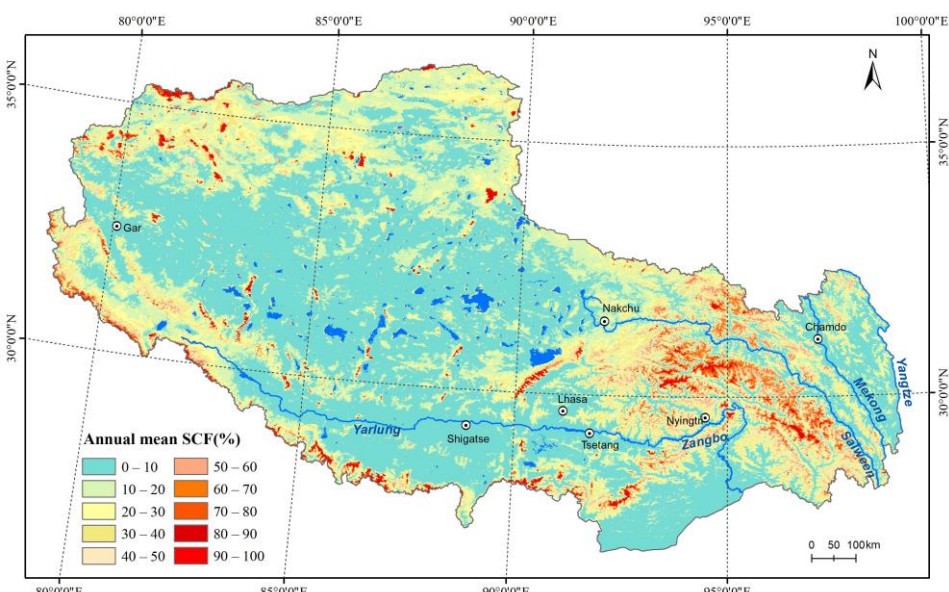

**Figure 2.** Annual mean SCF in Tibet.

In addition to terrain conditions, snow cover distribution in Tibet is strongly linked to the moisture availability and related atmospheric circulation systems over the TP. The rich snow cover in the Nyainqentanglha and its southeastern mountain regions is primarily caused by warm and moist air from the south driven by southwesterly flow [47], while high SCF in the west is mainly affected by orographic uplifting and low-pressure systems related to westerly disturbance during winter and spring [1,48]. However, the two least-snow-covered regions in the northern and southern Tibet resulted from prevailing descending air current during the snow seasons [49] and the strong shielding effect from surrounding high mountains [46,48].

4.1.2. Mean SCF in Spring

The spatial distribution of snow cover in Tibet in spring is similar to annual mean snow cover, generally showing that Nyainqentanglha mountain and its southeastern extension have the most concentrated areas of snow cover in the plateau in spring. This is followed by the southern Himalayas, Karakoram and western Kunlun mountains, whereas snow cover is limited in the interior of the plateau except for the high mountains. Vast central northern Tibet and the Yarlung Zangbo river valley are the two regions with the least snow cover on average in Tibet, as shown in Figure 3a. However, it can be clearly seen from the figure that areas with high SCF in spring is significantly larger than the annual average. Areas with an annual mean SCF > 50% only account for 6.6% of the total area of the plateau, while it reaches 14.5% in spring. Since spring is the transition season from winter to summer for atmospheric circulation systems over the TP, with strengthening of warm moist airflow in the south, snowfall weather on the plateau increases considerably. Moreover, snow cover increase in spring starts from the south, where there are higher snow cover occurrences during spring. The mean SCF in spring is 21.6%, only second to the winter season. The area with mean SCF < 10% is 43.9% of total area of the plateau and its spatial distribution is basically consistent with the annual average. The area with mean SCF < 50% is 85.6%

of the total area, and an area with a mean SCF > 60% accounts for 11.0% of total area and is primarily distributed in the Nyainqentanglha mountain and its southeastern extension, and upper part of surrounding high mountains.

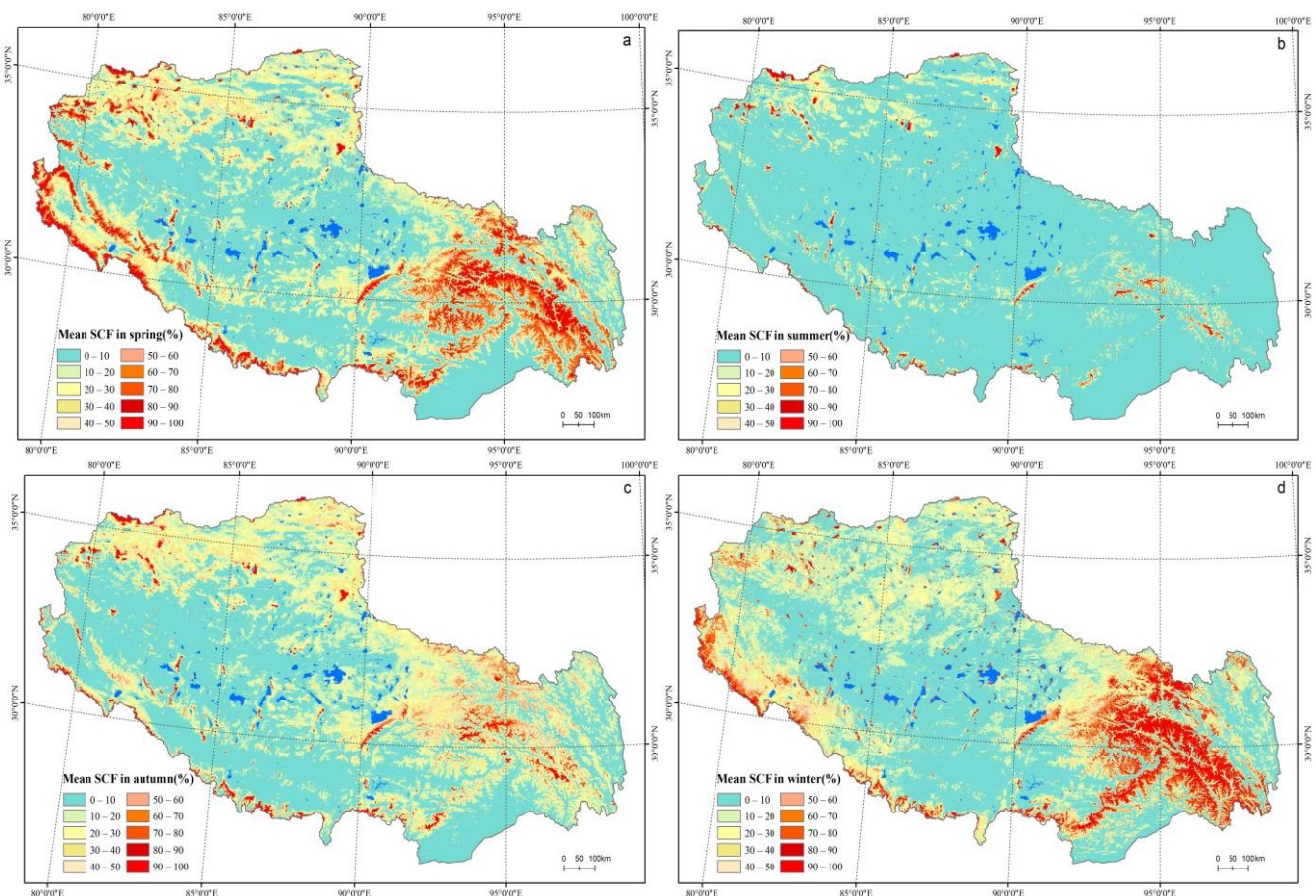

**Figure 3.** Seasonal mean SCF in Tibet in spring (**a**), summer (**b**), autumn (**c**) and winter (**d**).

### 4.1.3. Mean SCF in Summer

Figure 3b shows summer mean snow cover in Tibet from 2000 to 2014. It is clear that summer snow cover on the plateau is quite limited due to the temperature limitation and is only distributed in the upper part of high mountains in the interior and surroundings. Among these, the highest SCF is observed in the Kunlun Mountains in the north and Nyainqentanglha mountain and its southeastern extension, while there is by and large no snow cover distributed in the rest of broad area in Tibet. Summer mean SCF is 5.3% and the area with mean SCF < 10% accounts for 84.6% of total area. There is little difference between areas with a mean SCF < 50% and <60%, with roughly 98% of total area, and areas with a mean SCF > 60% are only 1.4% of the total area of Tibet. Summer is the rainy season in Tibet and the air temperature is above 0°C except in the high mountain ranges. Therefore, snow cover is very limited in vast areas of the plateau and is mainly distributed in the upper part of the interior and surrounding high mountains. Relatively, snow cover is rich in the Kunlun Mountains in the north and the Nyainqentanglha mountain range in the southeast.

### 4.1.4. Mean SCF in Autumn

Compared with the summer, the autumn mean SCF significantly increases, and its spatial distribution is generally consistent with annual and spring levels, which presents that higher mean SCF is in the interior alpine region and surrounding high mountains. Among these, the highest mean SCF is observed in the Nyainqentanglha and Tanggula

mountain regions and the northern plateau, whereas the least snow cover is found in the southern Tibetan valley and central part of northern Tibet, as shown in Figure 3c. In autumn, the mean SCF increase is obvious in the regions between the Nyainqentanglha and Tanggula mountains, and the northern plateau. However, compared with spring, in autumn the snow cover increase in the Karakoram and western Kunlun mountains in the north and in the southwestern Himalayas is not obvious, whereas the snow cover increase is obvious in broad alpine regions in the Nyainqentanglha, Tanggula and eastern Kunlun mountain ranges.

The autumn mean SCF is 16.2%, which is less than that in winter and spring, and almost equal to the annual average, mainly attributed to the lack of increase in snow cover in the west of the plateau during the autumn. In autumn, the area with snow cover less than 10% accounts for 48.3% of the total area, primarily including the Yarlung Zangbo river valley, the western part of the northern plateau and the southeastern river valleys. The area with a mean SCF < 50% is 94.2%, and that with a mean SCF < 60% accounts for 97.0% of Tibet's total area.

Located in the mid-low latitudes of the NH, autumn is transition season from summer to winter for atmospheric circulation systems over the TP. With the decrease in temperature and increasing activities of cold air from the north, favorable snow fall and snow cover conditions often result in larger snow cover on the plateau during the autumn, only second to winter and spring in terms of snow cover extent on average, and the snow cover increase in autumn generally starts in the north of the plateau.

### 4.1.5. Mean SCF in Winter

Figure 3d shows the winter mean SCF in Tibet from 2000 to 2014. The overall spatial distribution of snow cover is similar to those of the annual, spring and autumn mean snow cover, presenting that the highest snow cover is in the Nyainqentanglha mountain and its east, and southeastern part of Tanggula mountains, followed by the southwest and south of the Himalayas, whereas the lowest snow cover frequency is observed in southern Tibet and the central Qiangtang Plateau in the north. However, compared with those of spring and autumn, the winter snow cover increase is more obvious in the lower-latitude southeast and west. Snow cover increase is also obvious in the southeastern part of Tanggula Mountain.

The winter mean SCF is 21.7%, which is the highest among four seasons, with little difference from spring. The area with mean SCF < 10% is 38.8%, as the least in four seasons, which is mainly due to a more noticeable increase in snow cover in the northwest and south in the winter season. The area of mean SCF < 50% is 86.0% and the area of mean SCF < 60% is 89.1% of the total area. The area of mean SCF higher than 60% is 10.9%, principally distributed in Nyainqentanglha mountain, the southeastern part of Tangula mountain, and the Himalayan mountains in the south. A large spatial variation in snow cover exists in the Himalayan region due to the large climate and altitudinal gradients. Snow cover persistency shows a clear decreasing trend from west to east in all seasons and most of areas in the TP is snow-covered less than 20% of the total winter and spring period [50].

Snow cover distribution on the plateau during winter is not only affected by mountain topography, but also is closely related to moisture availability and atmospheric circulation systems over the plateau. During the winter season, the TP area is dominated by the westerlies and the large-scale vertical motion over the TP presents descending, which is unfavorable for snowfall. In contrast, a low temperature in winter is favorable for snow cover persistence on the surface. Furthermore, the southeast is often disturbed by warm and moist airflow induced by southwesterlies, which meets with cold air from the north, along with lower temperature in winter, creating weather and climate conditions conducive to snowfall and snow cover on the plateau [48,49].

### 4.2. Snow Cover Distribution with Elevation

Snow cover distribution in Tibet along with an elevation at a 1 km interval shows that the annual mean SCF in the two elevation zones below 2 km is very limited, with less

than 4% of the total area of Tibet. Snow cover then tends to increase with elevation and goes up to 22.3% at 5–6 km of altitude and 75.3% at above 6 km of altitude (Table 1). Some differences exist in seasonal snow cover distribution along an altitudinal gradient. In spring and autumn, a mean SCF below 2 km altitude is less than 3% and increases to 31.0% and 24.1% at 5–6 km altitude, respectively; the highest mean SCF is over 80% of the total area at above 6 km altitude, reaching 85.6% and 82.9%, respectively. In summer, the snow-covered area on the plateau is very small and is primarily distributed in the high-altitude area above 6 km, with a mean SCF of 71.6%, followed by the area between 5–6 km altitude with a mean SCF of 10.0%, and it is below 2.0% in the rest of the area. In winter, the highest SCF also occurs above 6 km altitude with 59.5%, but it is significantly lower than that in other seasons. At low altitudes, the snow cover increase with elevation in winter is more obvious than in other seasons, particularly at altitudes below 4 km.

**Table 1.** Annual and seasonal mean SCF with elevation zones in Tibet.

| No. | Elevation Range/km | Year/% | Spring/% | Summer% | Autumn/% | Winter/% |
|-----|--------------------|--------|----------|---------|----------|----------|
| 1 | <1 | 1.2 | 0.5 | 0.3 | 0.8 | 3.8 |
| 2 | 1–2 | 3.9 | 0.8 | 0.3 | 2.1 | 13.5 |
| 3 | 2–3 | 8.4 | 4.9 | 1.1 | 6.1 | 22.3 |
| 4 | 3–4 | 13.0 | 15.9 | 1.5 | 8.2 | 27.1 |
| 5 | 4–5 | 12.3 | 16.0 | 1.9 | 11.4 | 19.9 |
| 6 | 5–6 | 22.3 | 31.0 | 9.9 | 24.1 | 23.6 |
| 7 | >6 | 75.3 | 85.6 | 71.6 | 82.9 | 59.5 |

The more detailed spatial distribution of snow cover along with elevation in Tibet is further given in Figure 4, using a hypsographic curve based on an altitude interval of 500 m. Snow cover generally appears increase with elevation. The annual mean SCF increase is slow in areas below 5000 m altitude, with a mean SCF below 14%, while it increases rapidly at an altitude of 5000–7000 m, and the highest occurs around an altitude of 7000 m with 88.0%. Strong altitudinal dependence and a sharp increase in mean SCF between 5000 and 6500 m in elevation shown in hypsographic curve were observed in the Himalayan mountain region [23,24]. A high elevation in general terms means a lower temperature, which in turn leads to higher snow accumulation. The high mountain and low valley correspond well with more and less snow cover occurrences in Tibet.

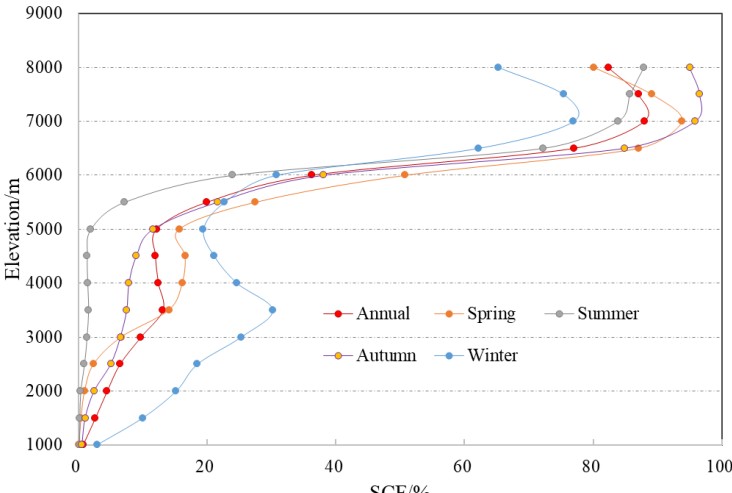

**Figure 4.** Annual and seasonal mean SCF with elevation in Tibet.

The seasonal mean SCF, except for winter, is less than 7% in areas below 3000 m and increasing trend with elevation is slow, whereas this trend tends to be obvious at an altitude of 3000–5000 m and becomes much more rapid at an altitude of 5000–7000 m. In terms

of altitudinal snow cover distribution in spring and autumn as the transition seasons, the mean SCF increase with an elevation below 3 km altitude is more obvious in autumn, while at altitudes of 3000–5000 m it is more obvious in spring, and at a high altitude over 7000 m the mean SCF in autumn is greater than that in spring. In winter, mean SCF in areas below 5000 m altitude is significantly higher than other seasons, which is closely linked to low temperature in winter, conducive to snow cover presence in the mountain regions. However, at higher altitudes above 6500 m, winter mean SCF is obviously less than other seasons. Seasonal mean SCF in Tibet generally reaches the maximum at an altitude of around 7000 m. Mean SCF at above 7000 m altitude is the highest in autumn, followed by spring and summer, while it is lowest in winter.

Figure 5 shows that monthly mean SCF in Tibet along with different elevation zones. Clearly, mean SCF in the area below 4 km altitude presents a unimodal pattern, where the higher the altitude, the higher the snow cover frequency, the more obvious the unimodal pattern and the longer the snow cover duration, as shown in Figure 5a. The peak value of mean SCF occurs in winter and its timing is delayed from January to February with the increase in altitude. The low temperature in winter plays an important role in the persistence of snow cover at lower altitudes. However, in the area above 4 km altitude, the intra-annual distribution of mean SCF presents a bimodal pattern, where the higher the altitude, the greater the mean SCF and the more obvious the bimodal pattern, and two peaks occur in spring and autumn, respectively. With the increase in elevation, the timing of the occurrence of the two peaks tends to be delayed from March in spring and advanced forward from November in autumn. The higher the altitude, the longer the snow cover duration in spring and autumn, which plays a more prominent role in sustaining snow cover on the plateau. At altitudes below 6 km, the lowest mean SCF occurs in summer, while at altitudes above 6 km it occurs in winter, which is closely related to the combined effect of atmospheric circulation, snowfall, temperature and topographic conditions in the study area. In the winter season, affected by cold high-pressure systems over the plateau driven by westerlies [51], the weather is fair and there is less snowfall, unless a large-scale disturbance occurs. Meanwhile, snow blowing due to strong prevailing westerly winds at higher altitudes accelerates snow loss [15]. Snow sublimation under strong insolation and dry weather further contributes to decreases in SCF during winter [17,51]. Sublimation contributes largely to decreases in SCF during the winter, especially for the areas with thin snow cover. More than half of snow mass was lost by sublimation in winter [18]. However, in the summer season the precipitation falls in the form of snow in the high-altitude regions and most of the continental glaciers in the TP are fed by snowfall during the summer season [48]. Therefore, the highest mean SCF above 6 km elevation occurs in summer instead of winter, similar to other seasons.

On a monthly average, the mean SCF in July is the lowest with 3.3%, and it reaches 18.8% in October. The highest mean SCF occurs in February with 24.9%, followed by March with 24.5%, while snow cover presents a decrease in December and January compared to November. Furthermore, among different elevations, the highest coefficient of variation(CV) of 1.4 occurs at 1000–2000 m altitude, indicating that monthly mean SCF variation is the largest in this region. CV progressively decreases with the increase in altitude and it decreases to 0.2 at above 6000 m, characterized by the higher the altitude, the greater the mean SCF and the lower the CV, indicating that the higher the altitude, the longer the snow cover duration, the more stable the intra-annual snow cover variations.

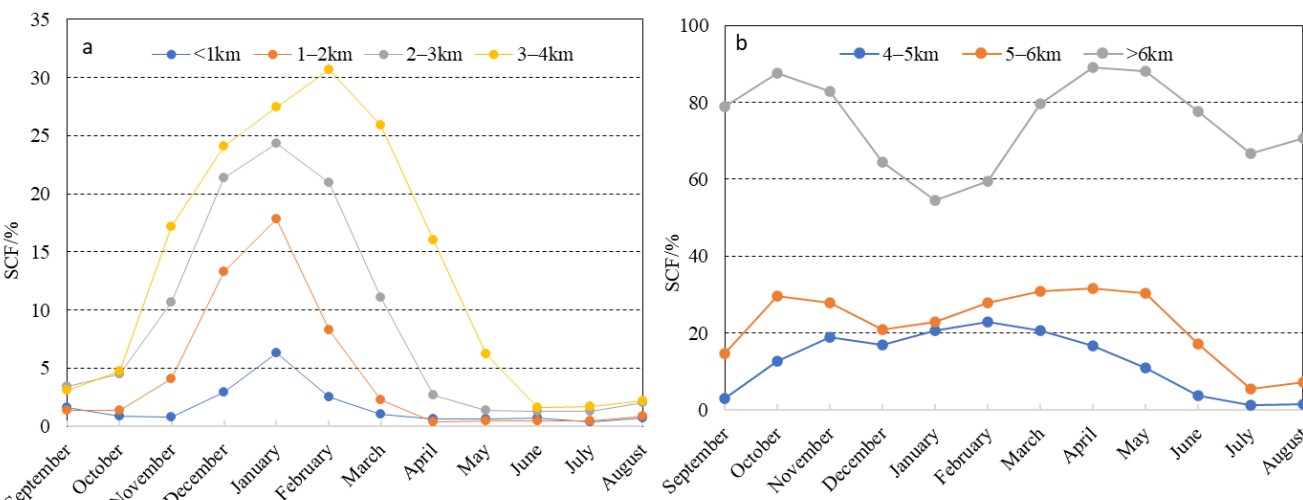

**Figure 5.** Monthly mean SCF with elevation zones in Tibet: below 4 km altitude (**a**) and above 4 km altitude (**b**).

### 4.3. Snow Cover Distribution with Longitude and Latitude

The geographical locations in the world represented by latitude and longitude are closely associated with solar radiation and water budget received on the ground, which in turn affects the spatial distribution of snow cover on the surface. The higher latitude receives less solar radiation and results in a lower temperature, which is more favorable for snow cover on the ground. The precipitation in Tibet presents decreases from southeast to northwest, which means that the southeast is more conducive to snowfall and snow cover in Tibet. However, the temperature decreases from south to north, implying the north is in favor of snow cover on the surface in terms of the temperature condition in Tibet.

The longitudinal snow cover distribution in Tibet is characterized by a high SCF in the western edge and east, and a low SCF in the vast central region and eastern edge, as shown in Figure 6. This is mainly due to high snow cover occurrences in the high mountain ranges, such as western Himalayas, Karakoram and western Kunlun mountains in the west, and Nyainqentanglha and its eastern extension, Tanggula mountain and eastern Himalayas in the east. A low SCF in the central region and eastern edge correspond with northern Tibet and southern Tibetan valleys, and eastern river valleys, where are the least snow covered areas in Tibet. In summer, snow cover on the plateau is very limited, with a mean longitudinal SCF of less than 9%. In autumn, the difference and variation of the mean SCF along with longitude is also small. However, winter and spring have a higher difference and variation for snow cover distribution from east to west. In spring, the mean SCF in the west is higher than that in winter, with smaller differences in central Tibet, whereas in the east, the mean SCF in winter is obviously higher than that in spring with the maximum difference occurring near 96°24′ E. The study shows that there is a clear decreasing trend in snow cover persistency from west to east in all seasons across the Hindu Kush Himalaya region [50].

Snow cover increase in Tibet with latitude is more obvious in the northern and southern margin, but this characteristic is not notable in the central plateau, as shown in Figure 7. A rapid increase in mean SCF with latitude in the southern edge is mainly related to altitudinal variations in snow cover from the southern slope of the Himalayas to the mountain peak between shorter horizonal distances. The obvious increase in snow cover with latitude in the north is due to more snow cover in the northern Kunlun mountains, while a low SCF near the latitude of 33°30′ N in central Tibet is the central Qiangtang Plateau, where is least covered by snow, apart from the valley regions in Tibet. At lower latitudes, the mean SCF in autumn is obviously lower than that in spring and winter, while at high latitudes it is obviously higher than that in the spring and winter. In winter, the mean SCF at low

latitudes is slightly higher than that in spring, but this difference is not obvious in other latitudinal zones.

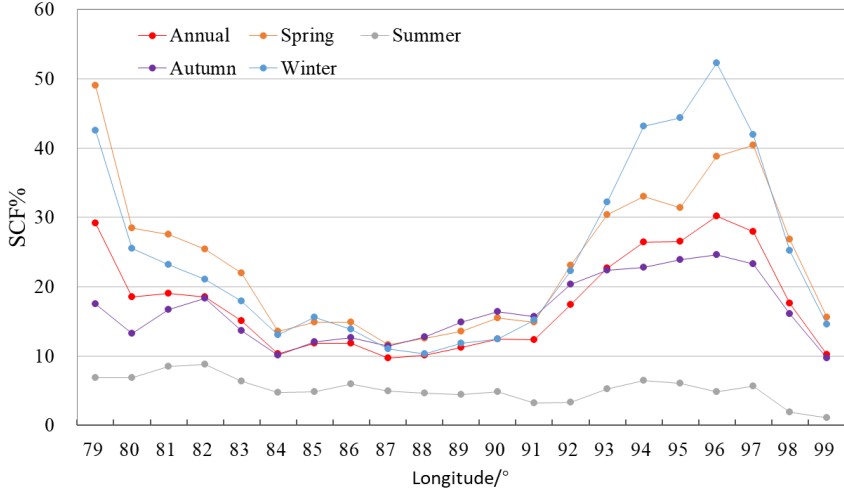

**Figure 6.** Annual and seasonal mean SCF with longitude in Tibet.

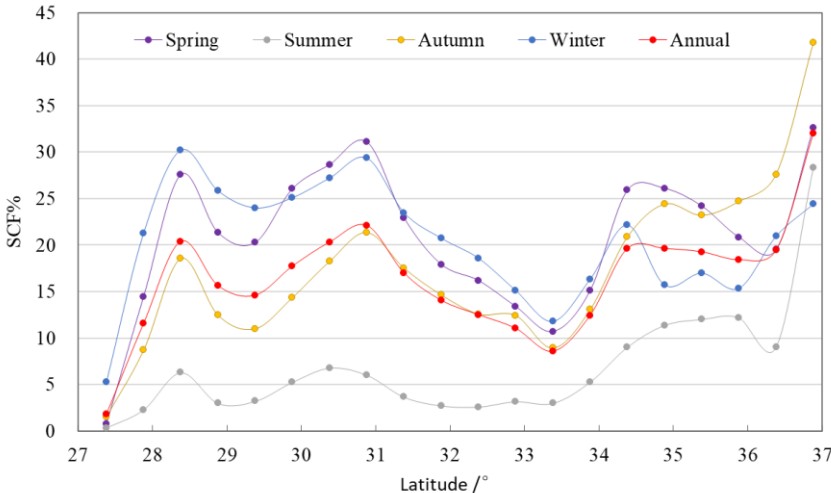

**Figure 7.** Annual and seasonal mean SCF with latitude in Tibet.

The obvious increase in snow cover with latitude in the north is due to more snow cover in the northern Kunlun Mountains, while a low SCF near 33°30′ N in the central Tibet is central Qiangtang Plateau, where is least covered by snow, apart from the valley regions in Tibet. At lower latitudes, the mean SCF in autumn is obviously lower than that in spring and winter, while at high latitudes it is obviously higher than that in spring and winter. In winter, mean SCF at low latitudes is slightly higher than that in spring, but this difference is not obvious in other latitudinal zones. Compared with elevation as main topographic factor, the dependence of SCF on the geographical latitude and longitude in Tibet is very weak.

### 4.4. Snow Cover Distribution with Aspect

The aspect, which represents the orientation of slope, is an important topographic factor that affects the spatial distribution of snow cover in mountain regions. Snow cover distribution on the different aspects in Tibet is shown in Table 2, demonstrating a bimodal distribution on four aspects within a yearly cycle. The mean SCF in January for the different aspects is above 17%, with the lowest value on the south-facing slope of 17.6% and the highest value on the north-facing slope of 24.9%. After January, snow cover increases on all aspects, and the monthly mean SCF reaches the first peak in February, except for the

eastward slope, while on the east-facing slope the first peak is observed in March, with 26.9%. From February to March, the mean SCF on the four aspects changes little, showing that the value on the south-facing slope is obviously lower than other three aspects. From March to May, the monthly mean SCF on the aspects starts to slowly decrease, while it decreases more rapidly after May and reaches the lowest level in July. The mean SCF in August increases slightly compared with July. With the decrease in temperature and end of the rainy season in the plateau, snow cover on different aspects shows a rapid increase from September and reaches a second peak in November, with the highest SCF of 26.5% on the north-facing slope and the lowest on the south-facing slope of 16.5%.

**Table 2.** Monthly mean SCF (%) with aspect zones in Tibet.

| No. | Aspect Range/° | January | February | March | April | May | June | July | August | September | October | November | December |
|---|---|---|---|---|---|---|---|---|---|---|---|---|---|
| 1 | −1 | 19.7 | 21.1 | 19.9 | 15.5 | 8.9 | 3.1 | 0.5 | 0.5 | 0.7 | 1.5 | 7.4 | 15.9 |
| 2 | 315–45° | 24.9 | 26.8 | 26.5 | 24.4 | 20.8 | 10.6 | 3.6 | 4.7 | 9.6 | 23.3 | 26.5 | 22.4 |
| 3 | 45–135° | 22.9 | 26.6 | 26.9 | 24.3 | 19.8 | 9.8 | 3.9 | 4.8 | 8.4 | 19.4 | 23.7 | 20.5 |
| 4 | 135–225° | 17.6 | 21.1 | 19.9 | 17.7 | 15.0 | 7.8 | 2.7 | 3.5 | 6.5 | 14.6 | 16.5 | 14.4 |
| 5 | 225–315° | 22.6 | 25.9 | 25.7 | 23.4 | 18.7 | 8.6 | 3.1 | 4.1 | 7.7 | 18.9 | 22.8 | 19.6 |

Snow cover on the flat terrain without orientation of the slope is obviously less than that with the aspects, and its intra-annual variation shows a symmetric bimodal distribution, being higher in winter, smaller in summer, and intermediate in spring and autumn. In the snow season, snow cover in Tibet starts to accumulate from October and reaches a peak in February, with smaller variations from January to March, while snow cover decreases rapidly from March and reaches its lowest average level in July or August. As discussed above, intra-annual snow cover distribution on different aspects in Tibet presents a bimodal pattern. The first peak occurs in February or March and the second peak is in November, which is agreement with overall snow cover distribution in the Tibetan Plateau. Likewise, the monthly mean SCF on the north-facing slope is the highest, while it is the lowest on the south-facing slope. With the exception that snow cover on the south-facing slope is obviously less than that on other aspects, there are no significant differences in snow cover frequency on the different aspects during the snow melting season after March, but significant differences exist in snow cover frequency on different aspects in the snow accumulation season in autumn.

Snow cover distribution on different aspects is closely linked with the redistribution effect of mountain aspects on hydrothermal conditions. The south-facing slope receives more solar radiation, causing higher temperature and more snow melt than on other aspects, whereas the north-facing slope receives much less solar radiation and low sublimation due to mountain shadows and has more persistent snow cover on the surface [16,29]. The significant impact of aspects on snow cover distribution is also observed in the trans-Himalayan region [24]. As north-facing slopes receive less solar radiation, snow accumulation starts earlier, and snow melt commences later in comparison to south-facing slopes. These characteristic differences indicate that high mountain topography affects the spatiotemporal distribution of snow cover via redistributing hydrothermal conditions and changing radiation balance in mountain regions.

*4.5. Snow Cover Distribution with Slope*

Based on the four slope zones below 5°, 5–10°, 10–20° and above 20°, monthly snow cover distribution on the topographic slopes was analyzed and is shown in Figure 8. The mean SCF on the slope below 5° is 17.0% in January, reaches its first peak of 18.8% in February, decreases slowly in spring, and reaches its lowest point of the year (2.0%) in July. The intra-annual variation of monthly snow cover follows a bimodal distribution, and its variation is relatively smooth compared to the slope zone above 5°. Snow cover on the slopes of 5–20° also presents a bimodal distribution. The mean SCF in winter and spring is obviously greater than that on the slopes below 5° and the timing of the peaks is delayed

by one month, occurring in March and November, respectively. The bimodal shape is also more noticeable than that on the slopes below 5°.

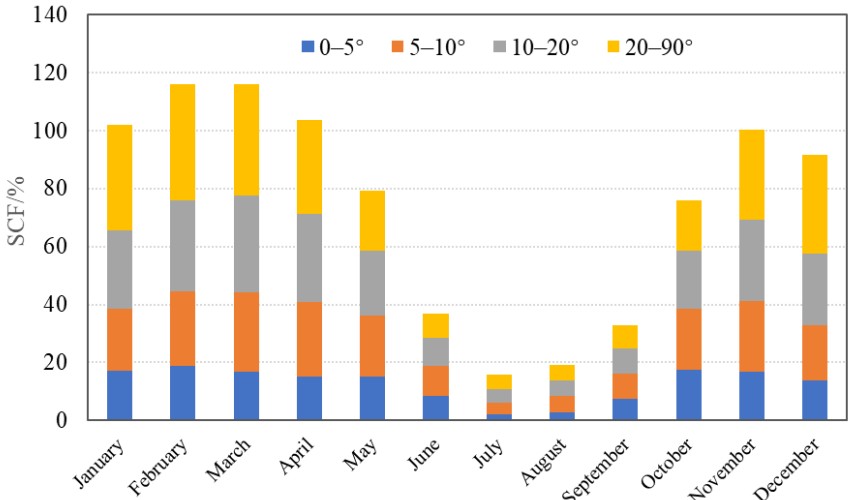

**Figure 8.** Monthly mean SCF with different slope zones in Tibet.

In the area with slopes above 20°, the mean SCF is 36.4% in January and reaches a peak of 40.1% in February, while the lowest SCF is in July, at only 12.5%. There is no distinct difference in snow cover distribution on the different slopes in the snow accumulation season before November, whereas the higher the slope, the more significant the snow cover frequency after November. In comparison with other slope zones, the mean SCF on slopes above 20° presents a unimodal distribution, with the peak value in February and lowest value in July. SCF shows that the higher the slope, the more abundant the snow cover from November to April, but this trend is not obvious on different slopes during other snow accumulation or melting periods. In addition, in terms of snow cover distribution on the plateau, the higher the slope, the faster the decline in mean SCF from the peak to the lowest level in July. There is no significant difference in the increasing rate of snow cover frequency on different slopes from July to November, but on slopes of less than 5°, it is slower than other slope zones. On the slopes below 20°, mean SCF shows a decreasing trend from November to December, followed by an increasing trend until the peak of snow cover in March.

To conclude, the intra-annual SCF on the slopes below 20° presents a bimodal pattern and two peaks on the slope below 5° appear in October and February, respectively, while the peaks on the slopes above 5° occur one month later, in November and March, respectively. The intra-annual SCF on the slope above 20° is unimodal and the peak appears in February. On the whole, for snow cover in Tibet, the higher the slope, the higher the occurrence of snow cover on the surface, and this spatial distribution is more obvious during the rich snow cover period from November to April. Furthermore, the higher the slope, the faster the decrease in snow cover from the peak in spring to the lowest level in summer. Meanwhile, there is no distinct difference in the increasing rate of snow cover on the different slopes at snow accumulation period from summer to autumn.

The annual mean SCF on the slope zone below 5° is the lowest (12.5%), showing that the higher the slope, the higher the frequency of snow cover. The mean SCF on the slope zone of 5–10° is 17.8%, while on the slope zones above 20°, it reaches 22.9% as the highest value (Table 3). At a seasonal level, the mean SCF below 5° slope is the lowest and shows that the higher the slope, the higher the snow cover frequency on the surface. The highest SCF in spring and winter occurs on the slopes above 20° (30.2% and 36.5%, respectively), while in autumn, it occurs on the slopes above 10° (19%). In summer, snow cover on the plateau is very limited, and the highest SCF occurs on the two slopes between 5 and 20° (6.4%). The increasing trend of snow cover along the slope is more obvious in winter.

The mean SCF is 16.3% on the slope below 5°, and on the slope zone above 20°, it reaches 36.5% as the highest during the four seasons. It can be concluded that the annual and seasonal mean SCF of slopes below 5° is the lowest in four slope zones, generally showing that the higher the slope, the higher the mean snow cover frequency, which is especially more obvious in winter and spring, and the highest mean SCF appears on the slope zone above 20° in winter. Snow cover distribution in mountain regions not only depends on the terrain height, but is also associated with slope gradient. The slope indicates the steepness of the terrain, usually shown in upward angles from the horizon. The mean SCF on the TP is the highest on the north-facing slopes, while the lowest mean is observed on the south-facing slope, and the SCF increases with the growth in steepness, as represented by the terrain slope [17].

**Table 3.** Annual and seasonal mean SCF with slope zones in Tibet.

| No. | Slope/° | Year/% | Spring/% | Summer/% | Autumn/% | Winter/% |
|-----|---------|--------|----------|----------|----------|----------|
| 1 | 0–5 | 12.5 | 15.6 | 4.3 | 13.8 | 16.3 |
| 2 | 5–10 | 17.8 | 24.8 | 6.4 | 18.0 | 21.8 |
| 3 | 10–20 | 20.4 | 28.6 | 6.4 | 18.8 | 27.5 |
| 4 | 20–90 | 22.9 | 30.2 | 6.0 | 18.7 | 36.5 |

## 5. Conclusions

High mountain topography is the main condition for snow cover persistence and accumulation in Tibet, but the relationships between topography and snow cover distribution in the Tibet area have not been well quantified until now. In this study, the spatial distribution of snow cover in Tibet and the impacts of topographic elements on snow cover distribution were quantitatively investigated based on the MODIS snow cover product and DEM data in this study. The main results are as follows:

(1) Snow cover in Tibet is very spatially uneven. There is generally more snow cover and high SCF on the Nyainqentanglha, western Gangdise, and surrounding high mountains, and less snow cover and low SCF in the southern Tibetan valley and central part of northern Tibet. Annual mean SCF is 16.3%, of which mean SCF in spring and winter is almost the same, with 22% each, followed by autumn (16.2%), and lowest in summer (5.3%).

(2) Snow cover in Tibet has a strong elevation dependence and is characterized by higher SCF corresponding well with high mountain ranges. The higher the altitude, the higher the snow cover frequencies, the longer the snow cover duration, and the more stable the intra-annual variation in snow cover. The mean SCF below 2 km altitude is less than 4%, while it reaches 75.3% at altitudes above 6 km. At altitudes below 6 km, the lowest mean SCF occurs in summer, while at altitudes above 6 km, it occurs in winter.

(3) Snow cover in Tibet generally increases with the mountain slope gradient; the higher the slope gradient, the higher the snow cover frequency. The south-facing aspect receives more solar radiation and stronger sublimation, causing less snow accumulation, while snow cover in north-facing areas receives less insolation and melts slower than on other aspects. The mean SCF on the north-facing slope is the highest and is the lowest on the south-facing slope. In comparison with topographic factors, the impact of geographical latitude and longitude on snow cover distribution in Tibet area is very limited.

(4) MODIS product v005 was used to present snow cover distribution and topographic dependence in Tibet in this study. At present, MODIS products v006 and v061 have been released to replace v005 with significant revisions. However, accurate evaluations of MODIS products (v006 and v061) in the TP area are still lacking. Our preliminary accuracy assessment shows that new MODIS products (v006 and v061) tend to overestimate snow cover on the TP, and it was found that many water bodies,

such as rivers in Tibet, are misclassified as snow cover pixels. Therefore, based on the accuracy evaluation of the latest MODIS snow cover products, more consistent and longer time-series snow cover products are expected to be developed for use in the future.

**Author Contributions:** D.C. processed data and wrote the manuscript; L.L. and Z.W. reviewed and edited the manuscript. All authors have read and agreed to the published version of the manuscript.

**Funding:** This study was financially supported by the Second Tibetan Plateau Scientific Expedition and Research (STEP) programme (2019QZKK010312;2019QZKK0603), the Independent Research Project of Science and Technology Innovation Base of Tibet Autonomous Region (XZ2021JR0001G), the Key Science and Technology Project of Tibet Autonomous Region (XZ202201ZD0005G01), and the National Natural Science Foundation of China (41561017).

**Institutional Review Board Statement:** Not applicable.

**Informed Consent Statement:** Not applicable.

**Data Availability Statement:** Not applicable.

**Acknowledgments:** The authors would like to acknowledge the U.S. National Snow and Ice Data Center (NSIDC) for providing the MODIS snow cover product (MOD10A2).

**Conflicts of Interest:** The authors declare no conflict of interest.

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
