# Peer review of "Spatial Distribution of Snow Cover in Tibet and Topographic Dependence"

_atmosphere, doi:10.3390/atmos14081284_

Round 1
Reviewer 1 Report
Spatial Distribution of Snow Cover on the Tibet and Topographic Dependence
This manuscript provides an interesting and important research topic, which has so far received less attention. While the paper clearly addresses research gaps, the authors should revise some sections before the paper can be published.
In the introduction (l 53-53), the authors refer to the importance of melt mater for local livelihoods. In this context I would recommend to cite papers that explicitly deal with the importance of meltwater for agricultural purposes and the potential social impacts of cryosphere changes. The authors should cite the following papers:
Mukherji et al. (2019): Contributions of the cryosphere to mountain communities in the Hindu Kush Himalaya: A review. Regional Environmental Change 19, 1311–1326. doi:10.1007/s10113-019-01484-w
Nusser et al. (2019): Cryosphere-fed irrigation networks in the north-western Himalaya: Precarious livelihoods and adaptation strategies under the impact of climate change. Mountain Research and Development 39(2), R1–R11. doi:10.1659/MRD-JOURNAL-D-18-00072.1
In line 56-58 the authors refer to the importance of topographical factors in the distribution of snow cover. They refer to articled from Tibet and Tienshan. In this context they should also cite the paper by Passang et al. (2022): Topographical impact on snow cover distribution in the Trans-Himalayan region of Ladakh. Geosciences 12(8), 311. doi:10.3390/geosciences12080311
The paper by Passang et al. also fits in the context of the larger HKH region referred to in lines 78-80.
As this paper also uses MODIS daily snow products, it would be interesting to compare results.
Line 95. It is not clear what the authors mean. “Tibet is main body of the TP and is located in southwestern part of TP…”. I do not understand the difference between Tibet and the Tibetan Plateau in the context of snow cover distribution. The authors explain later in the section “Study area” (line 109-110) “Tibet here refers to the Tibet Autonomous Region(TAR) administratively in China and is located in the southwestern part of the (TP)”. I think the authors should delete the sentence in line 95 and move it to the study area, where the context more clear.
Line 120: I would prefer to give rounded figures: “Annual precipitation ranges 70-883 mm …”. Rather say ranges from 70 to less than 900 mm. The very concrete figure 883 mm does not make sense here because the climatic station is not mentioned and in this overview context only the range is important.
The data and methods section is straight forward. Snow cover frequency is an appropriate approach. I only miss some words on cloud removal.
The authors have decided to present results and discussion in one section. The role of the higher ranges such as Nyainqentanglha has been clearly pointed out.
In figure 2 it might be good to insert at least the location of Lhasa or some main rivers or mountain ranges. Otherwise, readers who are not familiar with the region have to go back to Figure 1. This would not be necessary in Figure 3.
In the caption of Fig. 3 it would be good to name the months. The figure should even become clear before reading the main text. Summer (Jun – Aug), autumn (Sep – Nov), winter (Dec – Feb) and spring (Mar – May). Is this correct in your analysis? This was not explicitly mentioned.
I am not sure wether it is correct to use "the Tibet" in various sections of the manuscript, while it is certainly correct to use "the Tibetan Plateau" or the "Tibet Autonomous Region (TAR)".
Reviewer 2 Report
The high mountain topography is main condition that snow cover occurs and exists in the mid-low latitudes. This paper explained how mountain topographic elements (elevation, slope, and aspect) influences snow cover distribution on the Tibet area using MODIS snow cover product and DEM data. The paper is generally quite good from method, writing, to conclusion. I recommend the paper to be published in Atmosphere after some revisions. Some comments and revision points after reviewing the paper are as follows.
1. In line 32, SRTM in Keywords is suggested to be deleted.
2. In line 179, the full name of the SRTM should be given when it appears for the first time.
3. In line 67, sow cover should be snow cover.
4. It is suggested to modify Figure 1, where the hawkeye image covers the subject image. The hawkeye image should be shifted to the right.
5. In line 84-91, It is recommended to consider adding other relevant literature (as follows).
6. Table 1 and Figure 4 express similar meanings, it is recommended to delete Table 1
7. In global snow cover distribution, the latitude is very important factor affecting spatially snow cover distribution on the surface of Northern Hemisphere, generally showing latitudinal dependence in the NH. Why this pattern is not spatially obvious in Tibet area as shown in Figure
8. How to distinguish glacier from snow cover or perennial snow cover in this study since this area has largest ice mass and is known as “third pole”.
9. Some significant conclusions are drawn in this paper. However, it is suggested to discuss this important conclusion and the shortcomings of this paper. There is no discussion part in this paper.
Recommended relevant literature:
(1) Satellite observed spatiotemporal variability of snow cover and snow phenology over High Mountain Asia from 2002 to 2021. Journal of Hydrology,2022, 613, 128438.
(2) Spatiotemporal dynamics of snowline altitude and their responses to climate change in the Tienshan Mountains, Central Asia, During 2001–2019 [J]. Sustainability, 2021, 13(7): 3992.
(3)Spatiotemporal variation of snowline altitude at the end of melting season across High Mountain Asia, using MODIS snow cover product [J]. Advances in Space Research, 2020, 66(11): 2629-2645.
Reviewer 3 Report
(1) I suggest to revise the title as: Spatial Distribution and Topographic Dependence of Snow Cover on the Tibet.
(2) “A quantitative relationship between snow cover and topographic factors of the plateau has so far been lacking.” This is not right. There were many studies revealing the relation between snow cover and topography of the plateau. The results of this study should be compared and cross-validated with these studies.
(3) Fig. 1, the small map should be placed to the top-right corner of the big map.
(4) The condition “NDSI≥0.40” in the plateau tends to lead to under-estimation of snow cover, according to previous studies.
Extensive editing of English language required.
Round 2
Reviewer 1 Report
The authors have adequately responded to my earlier comments and have improved the manuscript significantly.
Reviewer 3 Report
The revision is fine. No comments added.